# A Residual N-Terminal Peptide Enhances Signaling of Depalmitoylated Hedgehog to the Patched Receptor

**DOI:** 10.3390/jdb12020011

**Published:** 2024-04-09

**Authors:** Sophia F. Ehlers, Dominique Manikowski, Georg Steffes, Kristina Ehring, Fabian Gude, Kay Grobe

**Affiliations:** 1Institute of Physiological Chemistry and Pathobiochemistry, Faculty of Medicine, University of Münster, Waldeyerstrasse 15, 48149 Münster, Germany; sophia.ehlers@outlook.com (S.F.E.); dominique.manikowski@medskin-suwelack.com (D.M.); ehring-kristina@web.de (K.E.); fgude@ucc.ie (F.G.); 2Institute for Neuro- and Behavioral Biology, Faculty of Biology, University of Münster, Röntgenstrasse 16, 48149 Münster, Germany; steff_00@uni-muenster.de

**Keywords:** hedgehog, patched, dispatched, cholesterol, palmitate, N-terminal peptide, signaling

## Abstract

During their biosynthesis, Sonic hedgehog (Shh) morphogens are covalently modified by cholesterol at the C-terminus and palmitate at the N-terminus. Although both lipids initially anchor Shh to the plasma membrane of producing cells, it later translocates to the extracellular compartment to direct developmental fates in cells expressing the Patched (Ptch) receptor. Possible release mechanisms for dually lipidated Hh/Shh into the extracellular compartment are currently under intense debate. In this paper, we describe the serum-dependent conversion of the dually lipidated cellular precursor into a soluble cholesteroylated variant (Shh^C^) during its release. Although Shh^C^ is formed in a Dispatched- and Scube2-dependent manner, suggesting the physiological relevance of the protein, the depalmitoylation of Shh^C^ during release is inconsistent with the previously postulated function of N-palmitate in Ptch receptor binding and signaling. Therefore, we analyzed the potency of Shh^C^ to induce Ptch-controlled target cell transcription and differentiation in Hh-sensitive reporter cells and in the *Drosophila* eye. In both experimental systems, we found that Shh^C^ was highly bioactive despite the absence of the N-palmitate. We also found that the artificial removal of N-terminal peptides longer than eight amino acids inactivated the depalmitoylated soluble proteins in vitro and in the developing *Drosophila* eye. These results demonstrate that N-depalmitoylated Shh^C^ requires an N-peptide of a defined minimum length for its signaling function to Ptch.

## 1. Introduction

The Hedgehog (Hh) family of morphogens is essential for animal development and plays a major role in stem cell maintenance and adult tissue regeneration [1,2,3,4,5]. Known activities of the vertebrate Hh ortholog Sonic hedgehog (Shh) include patterning of the embryonic ventral neural tube and posterior limb bud, the somites, brain, foregut, lung, and face (reviewed in [6]). Paracrine signaling in Shh-expressing cancers is also well established [7]. In *Drosophila*, Hh acts as a signaling molecule during embryonic development and later in the imaginal eye discs, which can therefore be used as a robust and reliable readout of Hh biofunction. Hhs also control many other important processes in development and in cancer, including cell proliferation and migration. The underlying mechanisms that allow such functional versatility are poorly understood, but are likely to involve a highly conserved set of unusual post-translational modifications during Hh biosynthesis in the ER and in the Golgi apparatus. The first unusual modification of all vertebrate and invertebrate Hh family members begins with the autocatalytic cleavage of a 45 kDa precursor molecule by its C-terminal cholesterol transferase domain [8]. This reaction yields 19 kDa cholesteroylated vertebrate Shh and *Drosophila* Hh. Next, a membrane-bound acyltransferase (called Hh acyltransferase (Hhat) in vertebrates or Skinny Hedgehog (Ski) in invertebrates) adds a palmitate to conserved N-terminal cysteines (C85 in fly Hh or C25 in mouse Shh) [9,10,11]. A well-established role for both lipids is to attach nascent vertebrate and invertebrate Hhs to the outer plasma membrane leaflet of producing cells (Figure 1). However, it is less clear whether the well-established release factors Dispatched (Disp, a 12-pass transmembrane protein [12,13]) and Scube2 (signal peptide, cubulin domain, epidermal growth factor-like protein 2, a soluble glycoprotein expressed only in vertebrates [14,15]) remove Hhs together with both lipid anchors from the plasma membrane (Figure 1A), or whether Hh release requires proteolytic processing of the lipidated terminal peptides to render the Hh/Shh signaling domain soluble (Figure 1B). To address this question, previous studies have constructed, expressed, and analyzed artificial Hh proteins lacking one or both terminal lipids. These studies have shown that the replacement of the N-terminal cysteine acceptor with serine or alanine (C25→A/S in ^C25A/S^Shh, C85→A/S in *Drosophila* ^C85A/S^Hh) specifically blocks palmitate addition and impairs Hh biofunction in vitro and in vivo [9,10,16,17,18,19,20], although to varying degrees [17,18,20]. The current interpretation of these published results is that the retained N-terminal Hh lipid directly contributes to maximal Hh/Shh signaling to the Ptch receptor and that the solubilized protein always remains dually lipidated [21,22,23,24]. 

However, another possibility to explain the impaired biofunction of ^C25A/S^Shh and ^C85A/S^Hh is that the lack of N-palmitoylation during biosynthesis loosens Hh tethering to the plasma membrane of producing cells and thereby perturbs Disp- and Scube2-mediated Hh solubilization in an unpredictable manner. Conceivably, the lack of spatiotemporal control of Hh solubilization could also lead to variably impaired Hh biofunction, as observed [9,10,16,17,18,19,20]. In this study, we analyzed Shh solubilization using cell culture conditions that mimic physiological conditions in fly hemolymph or vertebrate interstitial fluid. To this end, we expressed dually lipidated Shh in the presence of 10% serum. Using reverse-phase (RP) high-performance liquid chromatography (HPLC) to compare proteins based on their hydrophobicity, we identified high levels of soluble Shh variants that retained their C-cholesteroylated peptide but lacked their N-terminal palmitoylated peptides as a consequence of proteolytic processing at the plasma membrane [25]. Throughout this study, we refer to this monolipidated serum-released Hh variant as Hh^C^/Shh^C^. Importantly, despite the observed N-processing and depalmitoylation during its release, Shh^C^ induced Shh target gene transcription and differentiation in Hh reporter cells. Consistent with this result, engineered, unpalmitoylated ^C25S^Shh^C^ induced similar transcriptional and differentiation profiles when compared to Shh^C^ in vitro. Both results demonstrate that the N-terminal lipid is not essential for Ptch receptor activation in the presence of serum and that the dual-lipidated cellular Hh/Shh precursor may undergo different post-translational modifications upon release to enhance the versatility of Hh signaling. We also observed that the additional deletion of N-terminal amino acids from the unpalmitoylated protein abolished most of its biofunction in vitro and in vivo. This suggests that Hh^C^/Shh^C^ biofunction requires a short retained N-peptide, limiting proteolytic processing of the Shh N-terminus to sites most proximal to the plasma membrane.

## 2. Materials and Methods

### 2.1. Fly Lines

Ectopic expression of Hh variants or mCherry fusion proteins in the morphogenetic furrow of the eye disc was performed by crossing the following fly lines: UAS-Hh variant/Cherry*/CyO^WeeP^;hh^AC^/Tm6B* and *GMR-GAL4/GMR-Gal4;hh^bar3^/Tm6B*. The resulting *UAS*-Hh variant/Cherry*/GMR-GAL4;hh^bar3^/hh^AC^* flies were analyzed using a Nikon SMZ25 microscope (Nikon, Tokyo, Japan). *w^−^;+/+;hh^bar3^/hh^AC^* flies served as negative controls; *GMR-GAL4/+* flies served as positive controls. Hh constructs were inserted into the 51C1 landing site and mCherryC constructs into the 68E landing site (BestGene, Chino Hills, CA, USA) by using germline-specific PhiC31 integrase [26]. mCherryC on the third chromosome was first recombined with hh^AC^ before being crossed with *GMR-Gal4;hh^bar3^/Tm6B* to obtain *GMR-Gal4;hh^bar3^/UAS-mCherryC hh^AC^* flies. Flies were crossed at 18 °C to achieve moderate protein expression, following an established protocol [27].

### 2.2. Generation of Recombinant Proteins

Shh expression constructs were generated from murine cDNA (NM_009170: nucleotides 1–1314, corresponding to amino acids 1–438 and human Hhat cDNA (NM_018194). Both cDNAs were cloned into pIRES (Clontech) for their coupled expression from bicistronic mRNA to achieve near-quantitative Shh palmitoylation. Non-cholesteroylated ShhN (nucleotides 1–594, corresponding to amino acids 1-198) and non-palmitoylated ^C25A/S^Shh cDNA (amino acids 1–438) were generated by site-directed mutagenesis (Stratagene) and inserted into pcDNA3.1 (Invitrogen, Carlsbad, CA, USA). We also expressed N-truncated Shh versions and amino acid exchange versions as indicated. Co-expressed human Scube2 constructs were a kind gift from Ruey-Bing Yang (Academia Sinica, Teipei, Taiwan), furin, PACE4, and PCSK5/7 expressing cDNA were a kind gift from Nabil Seidah (Montreal Clinical Research Institute, Montréal, QC, Canada). Hh cDNA (nucleotides 1–1416, corresponding to amino acids 1–471 of *D. melanogaster* Hh) and HhN cDNA (nucleotides 1–771, corresponding to amino acids 1–257) were used. Two variants of the mCherry construct were inserted into the pUAST vector background and injected. Upstream of mCherry, the *Drosophila* Hh export signal sequence (codons 1 to 84) was followed by the *Drosophila* Hh extended N-terminal peptide with a functional palmitoylation acceptor cysteine (p, codons 85 to 98). C-terminal to the mCherry fluorophore, the three most C-terminal amino acids of the Hh signalling domain (amino acids VHG) were followed by the *Drosophila* Hh C-terminal domain (HhC), which is required for covalent cholesteroylation of the VHG glycine. HhC extended from codon 258 to 472, with codon 472 being the Hh stop codon. The resulting construct was named PmCherryC. The mCherry control lacked the Hh N-terminus. The constructs were inserted into the *Drosophila* 3L 68E1 landing site by BestGene (Chino Hills, CA, USA).

### 2.3. Protein Expression

Bosc23 cells (a HEK293 derivative (Research Resource Identifier (RRID): CVCL_4401) obtained from D. Robbins, University of Miami, Coral Gables, FL, USA) were seeded into six-well plates and transfected with 1 µg Shh constructs, with or without 0.5 µg Scube2 constructs, by using Polyfect (Qiagen, Hilden, Germany). Cells were grown for 2 days at 37 °C with 5% CO_2_ in Dulbecco’s modified Eagle medium (DMEM) containing 10% fetal calf serum (FCS) and penicillin–streptomycin (100 µg/mL). Shh-conditioned media (referred to as +10% FCS throughout this study) was aliquoted, shock frozen in liquid N_2_, and stored at −80 °C for subsequent use. To produce the media called -FCS in this work, cells were washed and serum-free DMEM added for 6 h, the DMEM harvested, centrifuged at 300× *g* for 10 min to remove debris, the batch aliquoted, shock frozen in liquid nitrogen, and stored at −80 °C. In some experiments, serum-free medium was supplemented with 600 µg/mL methyl-β-cyclodextrin (CD, a cyclic oligosaccharide with a hydrophobic core to accommodate cholesterol and extract the sterol from the plasma membrane) for 6 h before harvesting. For Western blot analysis and protein quantification, frozen aliquots were thawed on ice and samples without FCS were immediately incubated with 10% trichloroacetic acid (TCA) for 30 min on ice, followed by centrifugation at 13,000× *g* for 20 min to precipitate the proteins. For protein pulldown (PD) from serum-containing media, 40 μL of heparin–sepharose beads were added to Shh-containing media (500 μL media per PD) and incubated overnight on a rotator at 4 °C. All precipitates and pulldowns were analyzed by reducing SDS-PAGE and immunoblotting by using goat-α-Shh antibodies (AF464, R&D Systems, Minneapolis, MN, USA), followed by incubation with horseradish-peroxidase-conjugated secondary antibodies and chemiluminesce detection (Pierce, Carlsbad, CA, USA). Where indicated, known amounts of dual-lipidated, HEK293-derived human Shh (R&D Systems, 8908-SH) were used as a size control and to quantify Bosc23-expressed, TCA-precipitated proteins on the same blot. Proteins were quantified on Western blots using ImageJ (NIH, Bethesda, MD, USA) and the values obtained were used to adjust the flash-frozen protein amounts for subsequent assays, such as C3H10T1/2 differentiation, to ensure that similar amounts of proteins from the frozen batches were used in all subsequent assays. C3H10T1/2 cells (RRID: CVCL_0190) were obtained from ATCC (CCL-226). For 5E1 immunoprecipitation and quantification of ^C25S^Shh^C^ and its N-truncated variants, 10 µg of monoclonal 5E1 antibody (DSHB) was coupled to 2.5 mg Protein A–sepharose beads (Sigma, St. Louis, MO, USA) per immunoprecipitation (IP). Five hundred microlitres of medium per IP was incubated overnight on a rotator. S2 cells (RRID: CVCL_Z232) were cultured in Schneider’s medium (Invitrogen) supplemented with 10% fetal calf serum and 100 μg/mL penicillin–streptomycin. The cells were obtained from C. Klämbt, University of Münster, Münster, Germany. For visualization of mCherry on the surface of S2 cells (CVCL_Z232, cells were obtained from C. Klämbt, University of Münster, Germany), S2 cells were transfected with constructs encoding mCherry^C^ or PmCherry^C^ together with a vector encoding *actin*–GAL4 via Effectene (Qiagen) and cultured in Schneider’s medium for 48 h before fixation. Cells were fixed with 4% PFA and incubated with polyclonal anti-mCherry antibodies (Invitrogen PA5-34974) under non permeabilizing conditions. mCherry^C^ or PmCherry^C^ was visualized with secondary anti-rabbit IgG (Dianova) using a Zeiss LSM700 confocal microscope (Zeiss, Oberkochen, Germany). All cell lines were tested negative for mycoplasma.

### 2.4. Chromatography

Reverse-phase HPLC separation depends on the binding of molecules to immobilized hydrophobic ligands attached to the stationary phase and their elution in order of increasing molecular hydrophobicity. RP-HPLC was performed as described [10] using a C4-300 column (Tosoh, Tokyo, Japan) and an Äkta Basic P900 protein purifier (Cytiva, Uppsale, Sweden). Briefly, Bosc23 cells were transfected with expression plasmids for Shh and monolipidated or unlipidated mutants as described previously. Two days after transfection, cells were lysed in RIPA buffer containing a complete protease inhibitor cocktail (Roche, Basel, Switzerland) on ice, ultracentrifuged, and the soluble total cell extract was precipitated with acetone. Media obtained after Shh expression was treated in the same way. Protein precipitates were resuspended in 35 µL of (1,1,1,3,3,3)-hexafluoro-2-propanol and solubilized with 70 µL of 70% formic acid, followed by sonication. Reverse-phase HPLC was performed using a 0–70% acetonitrile/water gradient with 0.1% trifluoroacetic acid for 30 min at room temperature. Elution samples were vacuum-dried, resolubilized in reducing sample buffer, and analyzed by SDS-PAGE and immunoblotting. The signals were quantified using ImageJ (NIH, Bethesda, MD, USA). HEK293-derived human Shh (R&D Systems, 8908-SH) was used as a double-lipidated control protein.

### 2.5. Verification of Mouse Mesenchymal Stromal Cells

To confirm the multipotency of the reporter cells routinely used in our laboratory to quantify Shh biofunction, we used a mouse mesenchymal stem cell functional identification kit (R&D, SC010) according to the manufacturer’s instructions. Briefly, C3H10T1/2 [28] mesenchymal stromal cells were cultured in MEM basal medium supplemented with 10% FCS and antibiotics (for adipogenesis) or DMEM supplemented with 10% FCS and antibiotics (for osteogenesis and chondrogenesis), and differentiation was induced by media supplements to induce adipogenesis (hydrocortisone, isobutylmethylxanthine, and indomethacin), osteogenesis (ascorbate phosphate, β-glycerolphosphate, and BMP-2) or chondrogenesis (dexamethasone, ascorbate phosphate, proline, pyruvate, and TGF-β3). Cells were cultured for 14 days (adipocytes and osteocytes) and 21 days (chondrocytes), with old media removed and fresh media added every 3–4 days, fixed, and the differentiation status confirmed using goat anti-mouse FABP-4 polyclonal antibody (adipocytes), sheep anti-mouse collagen II polyclonal antibody (chondrocytes), and goat anti-mouse osteopontin polyclonal antibody (osteocytes). Cells were stained with Alexa Fluor 488 NL-557-conjugated donkey anti-goat or anti-sheep IgG secondary antibodies and nuclei were counterstained with DAPI (blue) to confirm their multipotent status.

### 2.6. Shh Reporter Assays

C3H10T1/2 cells were grown in DMEM supplemented with 10% FCS and antibiotics. Twenty-four hours after seeding, Shh-conditioned media was mixed 1:1 with DMEM containing 10% FCS and antibiotics, and applied to C3H10T1/2 cells in 15 mm plates. In some cases, 0–6 ng of double-lipidated, HEK293-derived human Shh (R&D Systems, 8908-SH) served as positive activity controls during the assays. To some samples, 2.5 μM cyclopamine, a specific inhibitor of Shh signaling, and 1 µg/mL of Shh neutralizing antibody 5E1 [29] were added to confirm the specificity of the assay. In general, due to variable expression levels, mutant and wild-type proteins were adjusted to similar levels prior to the induction of C3H10T1/2 differentiation, as determined by Western blotting of an aliquot of the same stock. Cells were lysed 5–6 days after induction (20 mM Hepes, 150 mM NaCl, 0.5% TritonX-100, pH 7.4) and osteoblast-specific alkaline phosphatase (Alp1) activity was measured at 405 nm after the addition of 120 mM p-nitrophenol phosphate (Sigma) in 0.1 M glycine buffer, pH 9.5. Assays were always performed in triplicate. Cells were authenticated and tested negative for mycoplasma.

### 2.7. qRT-PCR

C3H10T1/2 cells were stimulated with recombinant Shh^C^/^C25A^Shh^C^ in triplicate and the media was changed every 3–4 days. TriZol reagent (Invitrogen) was used to extract RNA from C3H10T1/2 cells after 14 days. A first-strand DNA synthesis kit and random primers (Thermo, Schwerte, Germany) were used for cDNA synthesis before a control PCR was performed with murine β-actin primers. Amplification with Rotor-Gene SYBR-Green on a BioRad CFX 384 machine was performed in triplicate according to the manufacturer’s protocol. Primer sequences are provided in the Appendix A. Cq values of technical triplicates were averaged, the difference to β-actin mRNA levels calculated using the ∆∆Ct method, and the results expressed as the log2-fold change if compared to the internal control of C3H10T1/2 cells stimulated with mock-transfected media.

### 2.8. Confocal Microscopy of Drosophila Eye Discs

Eye discs were dissected from L3 larvae, fixed, permeabilized, and mounted. Samples were stained with α-Ci155 antibodies (2A1 rat, DSHB, 1:100, overnight) and Alexa647-conjugated goat-α-rat antibodies (1:300) (Invitrogen). Stage-matched discs were immunolabeled by using the same antibody batch and dilution always following the same procedure. Images were captured on an LSM 700 Zeiss confocal microscope using ZEN software (Version 3.6), always with the same settings. 

### 2.9. Bioanalytical and Statistical Analysis

All statistical analyses were performed in GraphPad Prism by using one-way analysis of variance tests (ANOVA) (parametric, post-test as indicated, confidence interval 95%). Ommatidia from male and female flies were counted and statistically analyzed in the same way. qRT-PCR results were analyzed and visualized in GraphPad Prism MacOS version 9.4.1.

### 2.10. Molecular Modeling

The structure of Ptch1 with double-lipidated Shh has been visualized using the PyMol molecular graphics system, version 2.3.0, from Schrödinger LLC, Boston, MA, USA.

## 3. Results

### 3.1. Serum Promotes the Release of Selectively Depalmitoylated Shh^C^ from the Dual-Lipidated Cellular Precursor

To date, most studies have examined Shh release from expressing cells in serum-free or serum-depleted media [30,31,32,33]. However, serum starvation is known to affect the amounts of secreted proteins and their secretion pathways [34]. Furthermore, in vivo, Shh-expressing cells are immersed in interstitial fluid that circulates between cells and tissues. Interstitial fluid is a filtrate of blood serum through the capillary walls and therefore contains >20 g/L of proteins and low-mass lipoprotein particles from the serum [35]. Therefore, to test Shh solubilization under experimental conditions that better mimic the cellular microenvironment in vivo, we expressed several Shh variants together with the physiological release factors Disp [12,13] and Scube2 [14,15] in the presence of 10% FCS. We then asked whether Shh is released with both lipids still present or whether the protein undergoes (partial) delipidation during its solubilization in the serum-containing medium. To this end, we compared secreted Shh from serum-containing media with the cellular precursor or with commercially available dual-lipidated control proteins by reverse-phase high performance liquid chromatography (RP-HPLC). RP-HPLC is a powerful technique for separating proteins based on their hydrophobicity. The instrument was first calibrated with commercially available recombinant human R&D 8908-SH Shh. This protein is produced by detergent extraction from the plasma membrane of transfected HEK293 cells and therefore represents the dual lipidated Shh pre-release state upstream of Disp and Scube2 function. As shown in Figure 2A, R&D 8908-SH eluted predominantly in the late fraction #37 (black arrowhead). In contrast, an artificial monolipidated (N-palmitoylated) ShhN standard elutes in the earlier fractions #27–29 (Figure 2B, red arrowhead) and monolipidated (C-cholesteroylated) ^C25S^Shh elutes predominantly in and around fraction #32 (Figure 2C, white arrowhead). Non-lipidated artificial ^C25S^ShhN elutes in the earliest fractions #26–28 from the C4 column (Figure 2D, green arrowhead). We used the same C4 column and buffer stocks for the subsequent RP-HPLC characterization of cellular dual-lipidated Shh precursors and soluble proteins generated from these precursors. 

As expected from its established dual lipidation during biosynthesis, we found that the majority of pre-release Shh from Bosc23 cell lysates eluted in fraction #37 (Figure 2E, black arrowhead), just like the dual lipidated R&D 8908-SH control. We also noticed a fraction of less hydrophobic cellular proteins (Figure 2E, fraction #33, white arrowhead) that eluted similarly to non-palmitoylated ^C25S^Shh control proteins. This fraction likely represents overexpressed recombinant proteins that escaped Hhat-mediated N-palmitoylation during biosynthesis. Importantly, the same shift in hydrophobicity was observed when dually lipidated Shh was released from producing cells into media containing 10% FCS (Figure 2F, white arrowhead). This result indicates that the N-palmitate, but not the C-cholesterol, is removed during the release process. We call the resulting soluble depalmitoylated variant Shh^C^ to distinguish it from the cellular, dually lipidated Shh precursor. Non-palmitoylated but cholesteroylated artificial ^C25S^Shh^C^ control proteins eluted in the same fractions from the C4 column (Figure 2G, white arrowhead), confirming selective delipidation of the Shh^C^ N-terminus during release.

Notably, the replacement of serum with 600 μg/mL of the pharmacological cholesterol chelator methyl-β-cyclodextrin (CD, an oligosaccharide that complexes and shields cholesterol, thereby rendering it soluble) solubilized Shh^C^ and ^C25A^Shh^C^ with their C-terminal cholesteroylated peptides intact, yet again with the N-terminal palmitate removed (Figure 2H,I, fractions #32–34 represent N-processed Shh^C^ and ^C25A^Shh^C^ (white arrowhead), Shh^C^ and ^C25A^Shh^C^ fractions #27–28 represent small amounts of dually delipidated soluble Shh [32,33,36]). Taken together, these results suggest that physiological or pharmacological sterol acceptors likely interact with the C-terminal cholesteroylated Shh peptide to render it soluble, which in turn may protect it during release and limit the proteolytic processing of Shh to the plasma membrane-anchored N-peptide [25]. 

### 3.2. Shh^C^ Induces In Vitro Differentiation of C3H10T1/2 and NIH3T3 Cells

As described previously, it has been established that Shh/Hh expression in the dual lipidated form is required for unimpaired Hh biofunction in vivo [17,37,38,39,40] and that unpalmitoylated ^C25S^Shh is less bioactive than Shh [9,17]. To test whether selectively N-terminally processed Shh^C^ released in the presence of serum is still functional, we used the Ptch1-expressing multipotent fibroblastic C3H10T1/2 cell line as a reporter. We first tested the multipotency of our C3H10T1/2 cell line to differentiate into osteoblasts [28], chondrocytes [41], or adipocytes [42]. To this end, C3H10T1/2 cells were cultured in the presence of adipogenic, chondrogenic, and osteogenic supplements for different time periods and their responsiveness was confirmed based on the phenotype and the expression of cell surface markers (as shown in [25]). We then expressed Shh in serum-containing media in the presence of the physiological release regulators Scube2 [15,43] and Disp [12,13], normalized the proteins by Western blotting (inset), and added similar protein amounts to the C3H10T1/2 cells. As a control for the possible saturation of Ptch1 receptors on these cells, we also added the proteins at 50% concentration. As shown in Figure 3A, both Shh^C^ and ^C25S^Shh^C^ induced robust and concentration-dependent differentiation of C3H10T1/2 progenitor cells into alkaline phosphatase-producing osteoblasts (Shh^C^ at 100%: 2.16 ± 0.1 arbitrary units (au) (*n* = 3) and ^C25S^Shh^C^: 2.67 ± 0.26 au (*n* = 3), *p* < 0.0001, Shh^C^ at 50%: 0.88 ± 0.14 au (*n* = 3) and ^C25S^Shh^C^: 0.87 ± 0.09 au (*n* = 3), *p* > 0.999). As a negative control, ^C25S^ShhN bioactivity was strongly reduced (0.17 ± 0.005 au (100% protein, *n* = 3) and 0.13 ± 0.04 au (50% protein, *n* = 3), similar to the the mock (empty vector) medium control (0.13 ± 0.02 au (*n* = 3)). We confirmed the bioactivity of ^C25S^Shh^C^ by quantitative reverse transcription–polymerase chain reaction (qPCR) analysis of Ptch1 and Gli1 expression in C3H10T1/2 cells (Figure 3B, left two graphs). Ptch1 is known to be upregulated by the ligand [44], and Gli1 is a zinc finger transcription factor that operates downstream of Ptch1 and that is also transcribed in an Shh-dependent manner [45]. As a positive control, we added R&D 8908-SH Shh (labeled 8908 in Figure 3B) to fresh serum-containing medium to compare its biofunction with that of solubilized N-truncated Shh^C^ and unpalmitoylated ^C25S^Shh^C^ (inset). Again, ^C25S^ShhN served as a negative control. qPCR confirmed similar increases in the mRNA expression of Ptch1 by 8908-SH (we observed a 2.7-fold transcriptional increase), Shh^C^ (a 2.6-fold increase), and ^C25S^Shh^C^ (a 2.1-fold increase, ^C25S^ShhN: a 0.5-fold increase). Gli1 mRNA expression increased by 5-fold, 4.8-fold, 4.6-fold, and 0.5-fold, respectively. This suggests that N-terminal delipidation of Shh^C^ does not reduce its biofunction below that of the dually lipidated R&D 8908-SH control, at least when Shh^C^ and ^C25S^Shh^C^ are expressed in the presence of 10% serum. We confirmed this finding in the Shh-responsive NIH3T3 cell line (Figure 3B, right graphs). Again, 8908-SH and ^C25S^Shh^C^ induced Ptch1 mRNA expression to a similar extent (8908-SH induced a 1.67 ± 0.001-fold transcriptional increase, ^C25S^Shh^C^ a 1.39 ± 0.2-fold increase, ^C25S^ShhN: a −0.003 ± 0.12-fold increase, and Shh^C^ a 2.27 ± 0.37-fold increase). Gli1 transcription increased 2.4 ± 0.001-fold (8908-SH), 1.78 ± 0.16-fold (^C25S^Shh^C^), 3.1 ± 0.49-fold (Shh^C^), and −0.11 ± 0.25-fold (^C25S^ShhN). It is important to note that widely used alternative assays, such as the NIH3T3-Light2 assay for Gli1 transcription, require greatly reduced serum levels and often measure the activity of proteins expressed in the absence of serum. Differences between our results and previously published results [9,17] may therefore be explained by the different conditions used for protein expression and activity determination.

We next used qPCR of Hh target gene expression in multipotent C3H10T1/2 cells to test whether equivalent amounts of soluble Shh^C^ and ^C25A^Shh^C^ in serum-containing media induce similar or different expression profiles in other differentiation pathways of these cells (Figure 3C). We found that ^C25A^Shh^C^ and Shh^C^ not only induced proportional transcriptional changes in key targets of Hh signaling (Ptch1, Gli1-3) as previously shown, but also did not differentially affect the expression of regulators of adipogenesis (Dlk1, Pparγ, Fabp4, Cfd, Dgat2), osteogenesis (alkaline phosphatase (Alp1), also Spp1, Bglap, Runx2), chondrogenesis (Sox9, Col2a1, Col10a1, Col1a1, Mmp3), and proliferation (Cdk9, Mki67). 

### 3.3. N-Terminal Amino Acids Contribute to Shh^C^ Biofunction In Vitro

Previously, it was shown that aberrant processing of the most N-terminal 11 Shh amino acids by furin-like proteases prevents truncated Shh binding to Ptch1 [46]. Furin (also called proprotein convertase subtilisin/kexin type (PCSK) 3 or PACE) is one of seven members of the PCSK family that cleave their substrates at single or paired basic residues, mostly in the secretory pathway. Other members of this family are PCSK5 (also called PC5), PCSK6 (also called Pace4), and PCSK7. Initially, we expected that their cleavage specificity for single or paired lysines or arginines would make these proteases strong candidates to cleave the Shh N-terminal peptide at the highly conserved, polybasic HS-binding site (the so-called Cardin–Weintraub (CW) sequence K-R-R-x-x-K-K [47], where x is any amino acid) (Figure 4A, labeled in green). We screened all PCSK family members for their ability to cleave the CW sequence and found that only PCSK7 processed this site (Figure 4A, bottom inset, the right lane shows processed Shh^C^). Furin cleaves at a site following the minimal motif R-x-x-R or at the preferred motif R-x-K/R-R in the secretory pathway. Therefore, a G-to-R exchange just upstream of the CW site (generating ^G32R^Shh^C^ and ^C25A;G32R^Shh^C^) renders the molecules susceptible to furin cleavage (Figure 4A, inset). To test whether furin- and PCSK7 cleavage inactivated the solubilized proteins, we incubated C3H10T1/2 cells with similar amounts of recombinant furin-cleaved ^G32R^Shh^C^ and ^C25A; G32R^Shh^C^ and PCSK7-cleaved ^C25A^Shh^C^ (Figure 4B). In this assay, Shh^C^ and ^C25S^Shh^C^ served as positive controls. Indeed, furin/PCSK7 processing downstream of the CW site strongly reduced or abolished Shh biofunction, as indicated by strongly reduced or abolished induction of Ptch1, Gli1, and Alp1 mRNA expression (Figure 4B) [46]. In some cases, the transcription of Hh target genes was even suppressed below unstimulated control (mock) levels (transcription of all three target genes was most strongly suppressed by ^G32R^Shh^C^ + furin). We conclude from these results that proteolytic removal of a significant portion of the Shh^C^ N-terminus abolishes Hh-induced C3H10T1/2 cell differentiation, even when the proteins were expressed in the presence of serum. Note that reduced protein function after processing was not entirely dependent on the N-palmitate, as cleavage also reduced the activitiy of ^C25A^Shh^C^ (as observed for (incompletely) furin-cleaved ^C25A; G32R^Shh^C^ and for ^C25A^Shh^C^ in the presence of PCSK7).

The observed differences in Shh^C^/^C25A^Shh^C^ biofunctions in the presence or absence of subtilisin/kexin-type convertases suggested to us that the N-terminal Shh^C^ peptide contributes to Ptch1 binding and signaling. Such functions of the Shh N-peptide are supported by two recent cryogenic electron microscopy (cryo-EM) structures [22,23] (Figure 4C). These structures suggest multiple hydrogen bonds and salt bridges between the N-terminal palmitoylated peptide and Ptch1 to connect the most N-terminal Shh amino acids P^27^, R^29^, G^30^, and F^31^ with residues N^802^, Y^804^, N^940^, and Y^1013^ of the second extracellular domain of Ptch1, and to connect more than 10 residues of the first extracellular domain of Ptch1 with the Shh residues G^32^, R^34^, R^35^, H^36^, and P^37^. To test the putative functional role of these interactions (independent of postulated palmitate functions in Hh signaling), we expressed ^C25S^Shh^C^ and unpalmitoylated variants that additionally lacked increasing numbers of N-terminal amino acids. All solubilized proteins were pulled down using heparin–sepharose to control for their comparable production and secretion (Figure 4D, top), and media aliquots containing 10% FCS were added to C3H10T1/2 progenitor cells to induce their Hh-dependent differentiation into osteoblasts (in this experiment, Alp1 protein biofunction again served as a functional readout for Shh-induced differentiation). As observed previously, Shh^C^ and ^C25S^Shh^C^ induced C3H10T1/2 differentiation into Alp1-producing osteoblasts to a similar extent (Figure 4D, bottom). The teratogen cyclopamine (CA) [48] and the monoclonal antibody 5E1, which blocks Shh/Ptch1 interactions [49], both inhibited Shh-induced C3H10T1/2 differentiation, demonstrating the specificity of the assay (Shh^C^: 1.5 arbitrary units (au) ± 0.13 au, Shh^C^ + 5E1: 0.26 ± 0.01 au, Shh^C^ + CA: 0.19 ± 0.003 au, *p* ≤ 0.0001, *n* = 3; ^C25S^Shh^C^: 1.71 ± 0.03 au, ^C25S^Shh^C^ + 5E1: 0.21 ± 0.01 au, ^C25S^Shh^C^ + CA: 0.18 ± 0.003 au, *p* ≤ 0.0001, *n* = 3). Negative control media obtained from mock transfected Bosc23 cells were also inactive, as expected (0.12 ± 0.001 au). However, C3H10T1/2 differentiation induced by truncated ^C25S^Shh^C^ variants was variable: While ^C25S;Δ26–31^Shh^C^, ^C25S;Δ26–32^Shh^C^ and ^C25S;Δ26–33^Shh^C^ remained bioactive (^C25S;Δ26–31^Shh^C^: 1.41 ± 0.05 au, *p* = 0.64 when compared to Shh^C^; ^C25S;Δ26–32^Shh^C^: 1.26 ± 0.08 au, *p* = 0.01 when compared to Shh^C^ and ^C25S;Δ26–33^Shh^C^: 1.65 ± 0.27 au, *p* = 0.31 when compared to Shh^C^), the bioactivities of all proteins that were truncated beyond ^C25S^Shh^C^ amino acid 33 were strongly reduced (^C25S;Δ26–34^Shh^C^: 0.7 ± 0.04 au; ^C25S;Δ26–35^Shh^C^: 0.38 ± 0.002 au; ^C25S;Δ26–36^Shh^C^: 0.39 ± 0.003 au; ^C25S;Δ26–37^Shh^C^: 0.46 ± 0.002 au; ^C25S;Δ26–38^Shh^C^: 0.36 ± 0.01 au, *n* = 3 and *p* < 0.0001 for all forms when compared to Shh^C^). These results suggest that N-terminal Shh peptide interactions with the first, but not with the second, extracellular domain of Ptch1 contribute to Shh^C^ biofunction (Figure 4E) and that Shh^C^ processing likely occurs upstream of the amino acids R34/R35 to maintain the bioactivity of the solubilized protein.

### 3.4. The N-Terminal Hh Peptide Contributes to Morphogenetic Furrow Progression in the Drosophila Eye Disc

We next tested whether required interactions between the N-terminal amino acids of the ligand and Ptch1 receptors are conserved between vertebrates and invertebrates, e.g., whether the Hh-N-terminal peptide also contributes to Hh biofunction in vivo. For this purpose, we used *Drosophila* eye development as a model. The *Drosophila* eye consists of a honeycomb matrix of photoreceptors (ommatidia) that develop in a wave of differentiation that moves from the posterior to the anterior of the eye disc, the so-called morphogenetic furrow (Figure 5A). Here, cells located anterior to the furrow respond to Hh secreted by cells posterior to the furrow by producing the same protein. This creates a cyclic Hh signaling mode that drives the furrow across the entire disc [50] and determines the number of ommatidia in the adult eye. As a genetic background, we analyzed *hh^bar3^* eye discs in trans with *hh^AC^*, a homozygous lethal null mutation. These combined mutations impair endogenous Hh production and furrow progression in the disc, resulting in small kidney-shaped eyes consisting of 100 ± 8 ommatidia/eye instead of 610 ± 18 ommatidia/eye in the wild type (*hh^bar3^*/*hh^AC^* versus GMR>, *n* = 16 male eyes were analyzed, Figure 5B,C) [51]. To restore the phenotype, we used an established eye disc-specific *GMR* (*glass multimer reporter*)-*Gal4* driver [52] to express Hh or Hh-GFP [53] to visualize the protein (Figure 5A, top), non-palmitoylated ^C85S^Hh, and truncated variants thereof from the same *attP 51C* landing site to ensure similar protein expression levels [26]. We found that *GMR-GAL4*-driven Hh expression in discs made deficient in endogenous Hh synthesis restored most eye development (676 ± 15 ommatidia/eye, positive control), whereas ^C85S^Hh restored a smaller part of the compound eye (325 ± 17 ommatidia/eye, *n* = 16, and 361 ± 25 ommatidia/eye, *n* = 31, respectively, Figure 5B,C). Mechanistically, Figure 5A suggests that the reduced number of ommatidia resulted from a delay in furrow progression across the disc: In stage-matched larvae, the furrow driven by ^C85S^Hh expresses Ci155 as a consequence of signaling activation in anterior receiving cells (bottom), demonstrating ^C85S^Hh biofunction (as previously observed for ^C25S^Shh (Figure 3 and Figure 4)), but it lags behind the furrow driven by Hh (shown in the middle)). An important finding in this system is that the additional deletion of six N-terminal amino acids (resulting in the N-terminal peptide S-G^92^-R^93^-H^94^-R^95^-A^96^-R^97^-Hh, Figure 5B) significantly increased ^C85S;∆85–91^Hh biofunction (423 ± 23 ommatidia, *n* = 16). This finding suggested to us the possibility of preferential physiological N-terminal processing at this site during Hh release. ^C85S;∆85–93^Hh and ^C85S; ∆85–94^Hh activities were similar to ^C85S^Hh (359 ± 24 ommatidia/eye and 352 ± 27 ommatidia/eye, respectively), as were ^C85S;∆85–95^Hh activities (306 ± 6 ommatidia/eye). However, additional truncations reduced the activities of the ^C85S^Hh variants to varying degrees (^C85S; ∆85–96^Hh: 198 ± 25 ommatidia/eye; ^C85S;∆85–97^Hh: 274 ± 21; ^C85S;∆85–98^Hh: 211 ± 20; ^C85S;∆85–99^Hh: 239 ± 25; ^C85S;∆85–100^Hh: 127 ± 16, *n* = 16 and *p* < 0.001 for all forms when compared to ^C85S^Hh), consistent with the abolished activities of extensively truncated ^C25S^Shh^C^ variants in vitro (Figure 4A). We note that the strong loss-of-function of truncated ^C85S^Hh variants was not caused by partial or complete deletion of the three basic CW amino acids present in fly Hh (Figure 5B, top) [54], because *GMR-GAL4*-driven expression of Hh^3xA^, which has this site functionally deleted, restored most eye development (619 ± 22 ommatidia/eye, *n* = 9). Taken together, our results show that the N-terminal peptide contributes to Hh biofunction in vitro and in vivo. This led us to ask whether short palmitoylated N-peptides alone could regulate Ptch function in vivo, as previously suggested [55].

### 3.5. Isolated Palmitoylated or Unpalmitoylated N-Terminal Peptides Are Not Active In Vivo

Using the same *Drosophila* eye developmental model, we tested whether isolated, palmitoylated, or unpalmitoylated N-terminal Hh peptides are sufficient for Ptch receptor binding and signaling in vivo. To this end, we replaced the N-terminal signaling domain of *Drosophila* Hh (HhN, amino acids 99–195, Figure 6A) with an mCherry tag (Figure 6B). This strategy ensured quantitative cholesteroylation of the C-terminal amino acid during mCherry secretion to the cell surface. We then generated two UAS-regulated expression constructs from the cholesteroylated mCherry backbone: One with its N-terminal Hh peptide palmitoylated by endogenous Ski palmitoyltransferase expression (termed PmCherry^C^), and an mCherry^C^ control lacking the N-terminal lipid and peptide (Figure 6B). Expression of both constructs under an actin–GAL4 control in S2 cells confirmed their similar biosynthesis, secretion, and cell surface association (Figure 6C). For subsequent in vivo assays, both constructs were inserted into the *attP 68E* landing site on the third fly chromosome and expressed under *GMR*-*GAL4* control in *hh^bar3^/hh^AC^* eye discs (Figure 6D,E). Again, Hh expression as a positive control restored eye development (855 ± 40 ommatidia/eye, *n* = 7; negative control *hh^bar3^/hh^AC^* eye discs: 98 ± 8 ommatidia/eye, *n* = 8; wild type: 876 ± 56 ommatidia/eye, *n* = 7). In contrast, both mCherry^C^ and PmCherry^C^ were both inactive (105 ± 18 ommatidia/eye, *n* = 28 and 100 ± 19 ommatidia/eye, *n* = 27). We conclude that interactions between the palmitoylated Hh N-peptide and the receptor alone are not sufficient to induce signaling and that physical linkage of the N-peptide to the globular Hh domain is a minimal requirement for the generation of bioactive protein, at least in the *Drosophila* eye model. However, we note that another explanation for our result is that the interactions of the N-peptide may have been blocked by the attached mCherry tag, a possibility that we cannot rule out.

## 4. Discussion

Recently published cryo-EM studies have suggested several possible modes of Shh binding to its receptor Ptch1—without lipids, with one lipid, or with both lipids involved. The latter possibility was supported by Qi et al. who reported high-resolution structures of Ptch1 together with dually lipidated R&D 8089-SH/CF [22,23]. In these structures (Figure 7A, shown in yellow; Figure 7B shows the Ptch1 surface), the palmitoylated Shh N-terminus inserts deep into a Ptch1 “conduit” to block Ptch1-mediated cholesterol transport, which is also thought to pass through this conduit [57]. This proposed mode of Ptch control may represent one point in a conceivable spectrum of Ptch activity. However, the regulation of Ptch1 activity by monolipidated Shh variants carrying only N-palmitate [58] or only C-cholesterol [25] has also been demonstrated. Support for these alternative signaling modes comes from the in vivo finding that the unpalmitoylated protein is active in developing tissues, such as the mouse limb bud, because it leads to a spectrum of phenotypes similar to that induced by Shh [17]. Based on these findings, dual- or monolipidated Shh signaling to Ptch1 is not mutually exclusive but may be used differently in different tissues and developmental contexts to increase the versatility of the Hh pathway. 

We support this hypothesis by demonstrating that the dual-lipidated, cell surface-associated precursor is converted into C-terminally monolipidated Shh^C^ by N-terminal proteolytic processing. Support for proteolytic processing during release is provided by the increased electrophoretic mobility of the solubilized protein, as determined by SDS-PAGE/immunoblotting, and its decreased hydrophobicity, as determined by RP-HPLC. The proteolytic conversion of Shh during release is physiologically important because it is strictly dependent on the co-expression of the Shh release proteins Disp and Scube2 [25] (supporting the release mechanism shown in Figure 1B). The physiological importance of the proteolytic conversion of Shh is further supported by the functionality of the solubilized protein product in vitro and in vivo, although we note that the bioactivity of unpalmitoylated proteins was somewhat reduced in NIH3T3 cells and in the *Drosophila* eye disc. Our results show that this residual activity depends on Shh^C^ binding to Ptch1 via a protein–protein interface together with the N-terminal peptide to enhance signaling (Figure 7C). However, in contrast to the amino acids proximal to the globular domain, the most N-terminal amino acids of Shh^C^ (Figure 7D, shown in red) are not essential for the regulation of Ptch1 activity (Figure 7E). This result is consistent with the published finding that furin-mediated removal of eleven amino acids from the N-terminus of a mutated Shh, including amino acids that we determined to be critical for receptor activation, severely reduces Ptch binding and signaling [46]. The result also suggests that furin resistance of the wild-type N-terminal CW sequence, although rich in basic amino acids, is critical for protecting Shh during its secretion to the plasma membrane and during furin-mediated Disp activation at the cell surface [59,60] to maintain its biofunction.

We also show that Shh^C^ release from the plasma membrane requires the presence of serum. This result supports a model in which lipoproteins serve as soluble vehicles for lipid-linked morphogens in vitro and in vivo [58,61]. Notably, the retained presence of the C-cholesterol moiety in serum-released Shh^C^ is consistent with the proposed molecular mechanism by which Disp is thought to release Hh/Shh from the plasma membrane of producing cells. It is well known that all vertebrate and invertebrate Disp family members contain a sterol sensing domain (SSD) that is conserved in proteins that bind, transport, or respond to cellular sterols, such as SREBP cleavage activating protein (SCAP) and NPC1 [62,63,64]. The SSD extends into a hydrophobic surface channel that has been proposed to function as an open “slide” for lipophiles [65] for subsequent transfer to an acceptor [59]. This acceptor is likely to be a serum lipoprotein, as supported by the published concept that the fly lipoprotein called lipophorin carries cholesteroylated Hh in vivo [58,61,66] and a previous report showing that C-cholesterol is necessary and sufficient for Disp-mediated protein export [30]. Finally, our results show that the transfer is terminated by proteolytic processing of the palmitoylated N-peptide at a membrane-proximal position, allowing the globular domain of Hh/Shh and the N-terminal “stub” of lipoprotein-associated Hh^C^/Shh^C^ to interact with the receptor Ptch on the surface of the signal-receiving cell (Figure 7E).

## Figures and Tables

**Figure 1 jdb-12-00011-f001:**
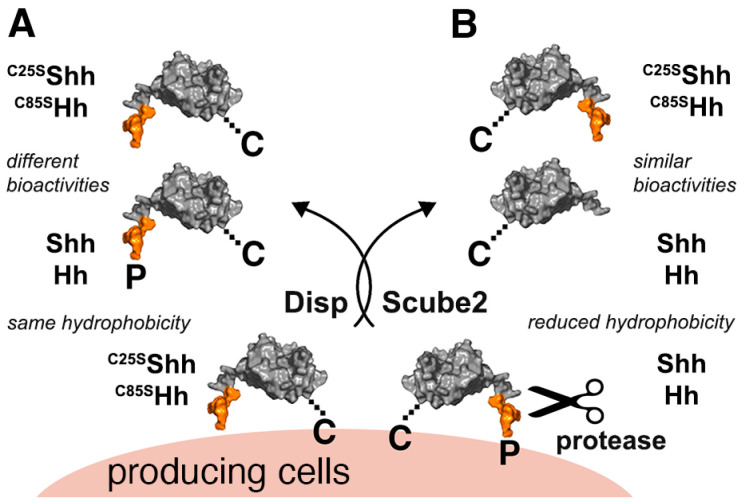
Schematic of Shh release from vertebrate cells or Hh release from invertebrate cells. All vertebrate and invertebrate Hh family members are synthesized as dual-lipidated proteins (N-terminally linked to pamitate (P) and C-terminally linked to cholesterol (C)) that are tightly bound to the outer plasma membrane leaflet of the producing cell. (**A**) One model proposes that the 12-pass transmembrane protein Disp delivers double-lipidated Shh and Hh to an acceptor (possibly Scube2). In this case, the hydrophobicity of the cellular protein and the solubilized protein should be similar. (**B**) Another model proposes that Hh/Shh is released by proteolytic processing (scissors) of the N-palmitoylated N-terminal peptide (shown in orange). This results in the solubilization of N-truncated monolipidated proteins. In this case, the hydrophobicity of the solubilized protein and the cellular precursor should be different. Furthermore, it has been shown that the replacement of the N-terminal cysteine acceptor with serine or alanine (C25→A/S in ^C25A/S^Shh, C85→A/S in *Drosophila* ^C85A/S^Hh) specifically blocks palmitate addition and impairs Hh biofunction, either by reducing ^C25A/S^Shh/^C85A/S^Hh binding and signaling to the receptor Ptch on recipient cells, or by circumventing the spatiotemporal control of monolipidated protein processing at the cell surface. The present study investigates these two possibilities: Highly disparate bioactivities between similar amounts of soluble Shh/Hh and non-palmitoylated ^C25A/S^Shh/^C85A/S^Hh would support a palmitate requirement for Ptch receptor binding and signaling (**A**), and similar bioactivities of both protein variants would not support this and suggest a different role for the N-linked palmitate in Shh/Hh function (**B**).

**Figure 2 jdb-12-00011-f002:**
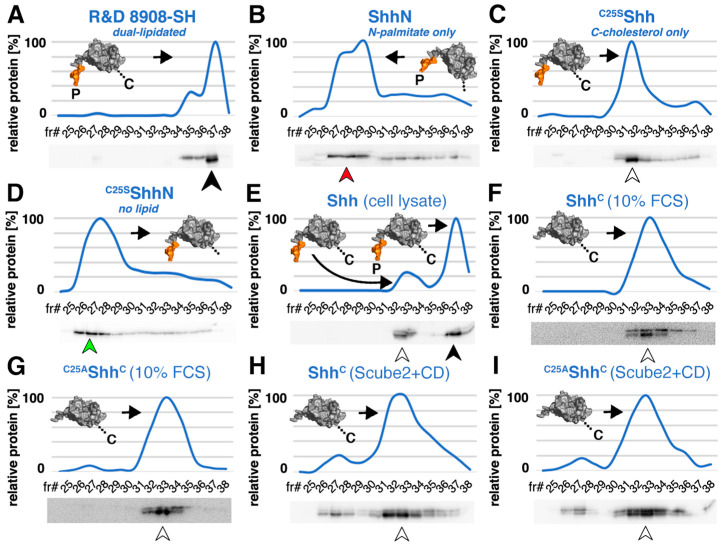
Reverse-phase HPLC reveals the decreased hydrophobicity of Shh^C^ released in the presence of serum or of pharmacological cholesterol acceptors. Goat-α-Shh antibodies (R&D Systems, AF464) were used for all blots to detect full-length unprocessed Shh (decreased electrophoretic mobility band, top) and N-truncated proteins that were solubilized from the cellular precursor (increased electrophoretic mobility band, bottom). Lower fraction numbers (fr#) indicate more hydrophilic (delipidated) proteins and higher fraction numbers indicate more lipophilic (lipidated) proteins. (**A**–**D**) RP-HPLC calibration. Consistent with its dual lipidation, R&D 8908-SH positive control proteins elute predominantly in the late fraction #37 (black arrowhead). Artificial monolipidated cellular ShhN elutes in fractions #27–29 (red arrowhead) and monolipidated cellular ^C25S^Shh (this artificial variant has the cysteine palmitate acceptor replaced with a non-accepting serine) elutes predominantly in fraction #32 (white arrowhead). Overexpressed soluble ^C25S^ShhN, another engineered control protein lacking both lipids, elutes in fractions #26–28 from the C4 column (green arrowhead). N-terminal Shh peptides in the schematics are labeled in orange. (**E**) Overexpressed cellular Shh elutes predominantly in fraction #37 (black arrowhead); a small fraction that probably did not undergo Hhat-dependent N-terminal palmitoylation elutes in fraction #33 (white arrowhead). (**F**) Shh^C^, solubilized by Disp and Scube2 from its dually lipidated cellular precursor (**E**), also eluted in fractions #32–34, showing that it retained the C-cholesterol moiety but not the N-palmitate after its release (white arrowhead). (**G**) Consistent with this, the artificially produced soluble control ^C25S^Shh^C^, blocked in its ability to undergo N-palmitoylation during biosynthesis, also eluted in fractions #32–34 (white arrowhead). The increased electrophoretic mobility of the protein indicated that its N-terminus was also processed. (**H**,**I**) Similar hydrophobicity of Shh^C^ and ^C25S^Shh^C^ was expressed in the presence of 600 μg/mL of the pharmacological cholesterol acceptor CD. Note that the electrophoretic mobility of the most soluble Shh^C^ is again increased (lower band), consistent with proteolytic processing of the palmitoylated N-terminal peptide during Disp- and Scube2-mediated Shh^C^ release [32,36].

**Figure 3 jdb-12-00011-f003:**
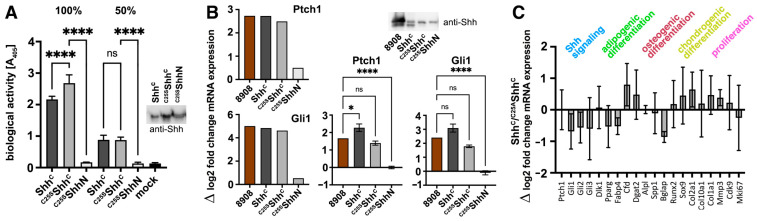
Similar activities of dual-lipidated Shh and depalmitoylated Shh^C^ variants. (**A**) Dual-lipidated Shh^C^, monolipidated ^C25S^Shh^C^, and unlipidated ^C25S^ShhN were solubilized from Disp-expressing Bosc23 cells in the presence of Scube2 and 10% FCS. The supernatant was first immunoblotted for quantification and protein normalization (inset). C3H10T1/2 cells were then stimulated with the same aliquoted protein batches at two different concentrations (100% and 50%) to control for the possibility of (over)saturation of the pathway. Data were normalized to mock-treated C3H10T1/2 cells. ****: *p* < 0.0001, n.s.: *p* > 0.999. (**B**) Immunoblotted proteins (inset) were used for quantification and normalization of Shh variant proteins and dually lipidated control R&D 8908-SH (inset, labelled brown in the graphs). All cholesteroylated proteins induced strong transcriptional increases of the Hh signaling targets Ptch1 and Gli1 in C3H10T1/2 reporter cells (left), and non-lipidated ^C25S^ShhN was inactive, as expected. Transcriptional increases are expressed as a log2-fold increase in mRNA expression normalized to mock-treated cells. Mean values of two independent experiments are shown. The same trend was observed in NIH3T3 cells (right). Averaged data from three independent experiments are shown. Ptch1: ****: *p* < 0.0001, *: *p* = 0.026, n.s.: *p* = 0.35. Gli1: ****: *p* < 0.0001, n.s.: *p* > 0.05. (**C**) Consistent with their similar hydrophobicity, N-processed Shh^C^ and unpalmitoylated ^C25S^Shh^C^ induced similar gene expression profiles in C3H10T1/2 reporter cells (all *p*-values are not significant, *p* > 0.04). These are expressed as the difference in log2-fold mRNA expression, normalized to mock-treated cells, between Shh^C^ and ^C25A^Shh^C^.

**Figure 4 jdb-12-00011-f004:**
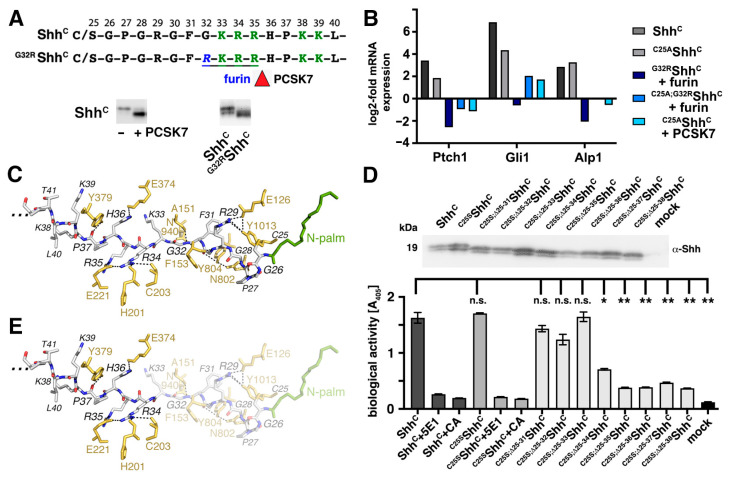
A minimal N-terminal amino acid sequence contributes to Shh signaling in vitro. (**A**) The N-terminal Shh peptide, including the Cardin–Weintraub (CW) motif (green) is shown. A G-to-R exchange (shown in blue) just upstream of the CW site renders ^G32R^Shh susceptible to furin cleavage (inset, right lane, red arrowhead). PCSK7 cleaves the wild-type Shh peptide at the same site (inset, and red arrowhead). (**B**) qPCR confirmed that Shh^C^ and ^C25A^Shh^C^ increased the transcription of Ptch1, Gli1, and Alp1 in C3H10T1/2 cells to a similar extent. Target gene transcription was much less induced by furin- or PCSK7-cleaved proteins, indicating that processing rendered them inactive. (**C**) Cryo-EM derived structures (pdb: 6e1h) reveal interactions between the palmitoylated Shh N-peptide (white backbone, nitrogens stained blue, oxygens stained red, palmitate stained green) and Ptch residues (stained yellow) [22]. (**D**) Shh and mutant proteins lacking the N-terminal C25 to prevent palmitoylation, and their consecutively N-truncated counterparts were expressed, pulled down with heparin, and immunoblotted. All proteins were expressed at similar levels, as indicated by polyclonal α-Shh reactivity. Bottom: C3H10T1/2 osteoblast progenitor cells were incubated with similar amounts of Shh^C^, ^C25S^Shh^C^, and their truncated variants, and relative increases in Alp1 activity were determined as biological readouts. Media obtained from mock-transfected Bosc23 cells was used as a negative control, and Shh^C^-conditioned media as a positive control. Inhibition of Shh^C^ signaling by 5E1 and cyclopamine (CA) confirmed the specificity of the assay. One-way ANOVA with Dunnett’s correction was used to determine the *p*-values. ** indicates significant activity if compared to Shh^C^ (*p* < 0.01). * *p* < 0.05. n.s. not significant (*p* > 0.05). (**E**) Loss of activity of ^C25S^Shh^C^ variants truncated beyond N-terminal amino acid 33 suggests that interactions of the Shh amino acids R34, R35, and H36 with the Ptch1 amino acids H201, C203, E221, E374, and Y379 are required for Ptch binding and signaling.

**Figure 5 jdb-12-00011-f005:**
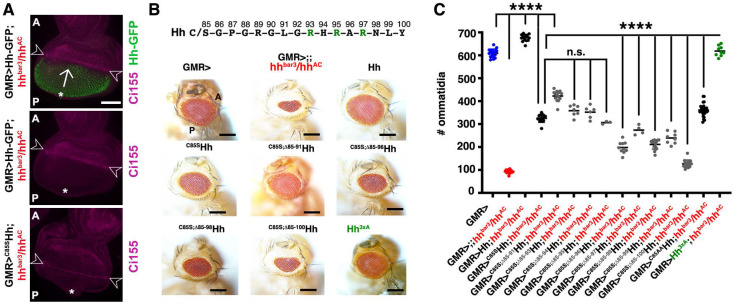
N-terminal ^C85S^Hh amino acids contribute to Hh-regulated eye development in *Drosophila*. (**A**) Top: In L3 larvae lacking most endogenous Hh expression (*hh^bar3^/hh^AC^*), Hh-GFP expression (green) under GMR > Gal4 control drives the morphogenetic furrow (Ci155-expressing line between arrowheads) from the posterior (P) disc tissue (asterisk) to the undifferentiated anterior (**A**) disc tissue (arrow). Middle: The same disc is shown without the green channel to better visualize the morphogenetic furrow (between the arrowheads). Bottom: GMR > Gal4-driven ^C85S^Hh expression in stage-matched discs also allows furrow formation and progression (Ci155-expressing line between the arrowheads), but to a more limited extent (as indicated by the reduced distance between the * and the furrow). Scale bar: 50 μm. (**B**) Top: The N-terminal peptide of *Drosophila* Hh. CW residues are shown in green. Bottom: Short-range Hh signaling in the eye disc determines the number of photoreceptors (ommatidia) in the adult *Drosophila* eye. GMR > transgenic flies were used as positive controls and GMR>;;hh^bar3^/Hh^AC^ flies that lack most endogenous Hh expression in the eye disc and that do not express hh transgenes served as a negative control. Such discs develop into very small eyes, but *GMR-GAL4*-controlled Hh or Hh^3xA^ (this form has all basic CW arginines (green) replaced by alanines) fully restore eye development in this background. *GMR-GAL4*-driven ^C85S^Hh, ^C85S;∆85–93^Hh, ^C85S;∆85–94^Hh, ^C85S;∆85–95^Hh, and ^C85AA^Hh expression in this background restored eye development to a limited extent (consistent with reduced furrow progression) and ^C85S;∆85–91^Hh expression restored eye development to a greater extent. However, additional deletion of N-terminal amino acids restored eye development to a much lesser extent. (**C**) Quantification of phenotypes shown in A. One-way ANOVA with Bonferroni correction was used to determine the *p*-values. ****: *p* < 0.0001, n.s.: not significant (*p* > 0.05). No significant sex differences were observed.

**Figure 6 jdb-12-00011-f006:**
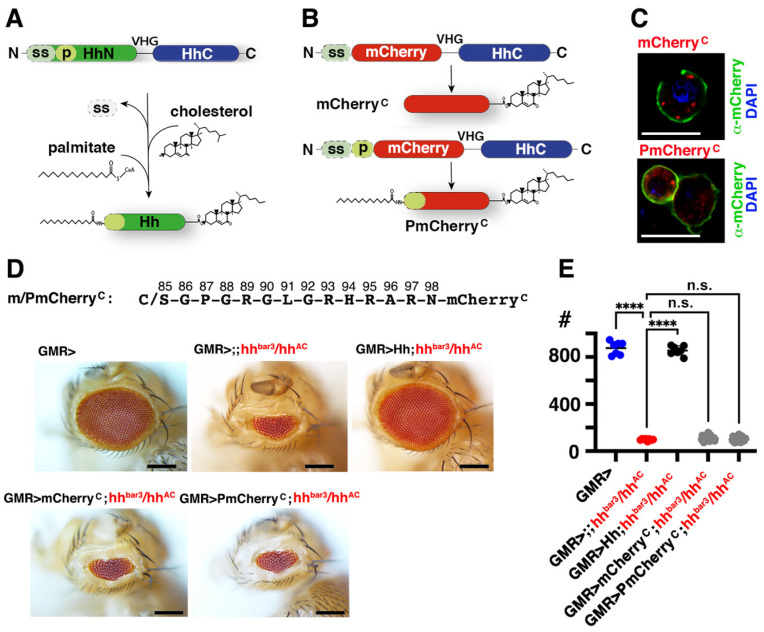
Isolated N-terminal Hh amino acids are not active. (**A**) Schematic of dual lipidated Hh/Shh production by autocatalytic cleavage of precursor proteins. Coupled precursor cleavage and cholesteroylation is mediated by the C-terminal cholesterol transferase domain (HhC), followed by palmitate attachment to the N-terminal cysteine by a membrane-bound acyltransferase to generate dual-lipidated Hh. ss: signal sequence for protein export into the ER, p: palmitoyltransferase recognition peptide [56]. VHG denotes the peptide sequence linking the domains. (**B**) HhN replacement by mCherry generates the cholesteroylated fluorophore. The presence of the palmitoyltransferase recognition peptide generates the dually lipidated fluorophore. (**C**) Monolipidated mCherryC and dual-lipidated PmCherryC are secreted to the outer plasma membrane leaflet of producing S2 cells. mCherry fluorescence is shown in red, α-mCherry antibody binding to non-permeabilized cells is shown in green, DAPI staining of the nucleus is shown in blue. Scale bars: 10 μm. (**D**) Top: The N-terminal Hh peptide attached to PmCherryC is shown. Below: Determination of protein activity in vivo. Eyes generated from discs lacking most Hh expression (*hh^bar3^/hh^AC^*) served as negative controls, and GMR > transgenic flies or *GMR-GAL4*-driven Hh expression in an *hh^bar3^/hh^AC^* background served as positive controls. *GMR-GAL4*-driven mCherryC or PmCherryC expression did not restore eye development. Scale bars: 100 μm. (**E**) Quantification of phenotypes. One-way ANOVA with Dunnett’s correction was used to determine the *p*-values. ****: *p* < 0.0001, n.s.: not significant (*p* > 0.05). No significant sex differences were observed.

**Figure 7 jdb-12-00011-f007:**
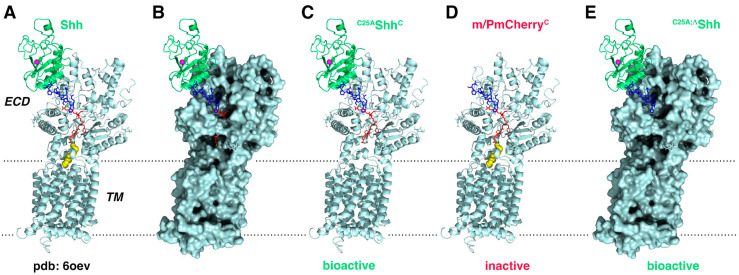
Models of Ptch activity modulation as tested in this study. Ptch receptors are shown in gray, the globular Shh domain is depicted in green, the Shh palmitoyl moiety in yellow spacefill, the N-peptide connecting the two in blue/red, and a zinc ion complexed by most vertebrate Hh family members in pink. (**A**) Pdb structure 6oev [23] shows a hydrophobic channel extending from the Ptch transmembrane (TM) region to the Shh protein-binding extracellular domain (ECD). Insertion of the palmitoylated Shh N-terminus is thought to regulate Ptch1 function. (**B**) Spacefill of the cartoon model shown in (A) to better illustrate the channel. (**C**) As shown in this study, deletion of the N-palmitate does not affect Ptch1 function when Shh^C^ is released in the presence of serum. (**D**) In contrast, the palmitoylated N-terminus alone is not bioactive. (**E**) Deletion of most of the Shh N-terminal peptide (red in A) does not affect protein function, but the additional deletion of N-terminal amino acids renders the truncated Shh protein inactive. This suggests that a minimal Shh/Hh N-terminal motif (labeled blue) contributes to the signaling process.

## Data Availability

The qPCR source data presented in this study are openly available at doi:10.7554/elife.86920.1., and other data and materials can be made available upon request to the corresponding author.

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
