# Peer review of "A Residual N-Terminal Peptide Enhances Signaling of Depalmitoylated Hedgehog to the Patched Receptor"

_jdb, 2024, doi:10.3390/jdb12020011_

Round 1

Reviewer 1 Report (Previous Reviewer 3)

Comments and Suggestions for Authors

The authors improved the manuscripts and study reasonably. The only thing I would like to say is:

'cloning of recombinant proteins" - proteins cannot be cloned. cloning of constructs or etc.

Comments on the Quality of English Language

N/A

Author Response

Reviewer: The authors improved the manuscripts and study reasonably. The only thing I would like to say is: “cloning of recombinant proteins” - proteins cannot be cloned.

Reply: Thank you for pointing out this error. The sentence now reads: "Production of recombinant proteins". (line 106).

Reviewer 2 Report (Previous Reviewer 2)

Comments and Suggestions for Authors

Review comments on revised submission:

I am very surprised that the authors just made minor changes/addition to the results and discussion. Critical questions exist.

1) Can ShhC be identified in the mouse model in terms of its activity in Shh signaling?

2) what is the in vivo activity of ShhC in the contest of shh mutation? This is critical because the ShhC might have functions toward the endogenous Shh protein.

3) how does the eight amino acids of ShhC contribute to its signaling activity? The authors do not carry out any experiments to address this question.

4) the Drosophila eye phenotypes need to be collaborated with immunostaining data to support the ideas.

Overall, the questions raised by this reviewer were not adequately addressed in this revision.

Author Response

This manuscript is a resubmission of an earlier submission. The following is a list of the peer review reports and author responses from that submission.

Round 1

Reviewer 1 Report

Comments and Suggestions for Authors

In this paper Ehlers et al ask the question of whether physical linkage of cholesterol SHH domain with a short N peptide independently of palmitoylation constitutes a minimal requirement for Ptch receptor binding and signaling. Their work focuses on the necessity of Sonic HedgeHog dual lipidation and its impact(s) on HH spatio-temporal solubility, diffusion and ability to activate Patch.

In this paper they show that in some biological context and in the presence of 10% serum, which mimics the in vivo conditions, Shh variants with their C-cholesteroylated peptide but no N-terminal palmitoylated peptide are secreted and soluble. Using various in vitro and in vivo assays they present evidence that this mono lipidated Shh form induces transcription of Shh target genes and differentiation of the C3H10T1/2 cells lines in vitro. Moreover, using an eye ommatidia assay they present data arguing that in certain in vivo contexts this form is active. 

More precisely their results indicate that:

1- The dual lipidated cell surface associated SHH precursor converts into C terminally mono lipidated form in the presence of serum. This conversion is physiologically important and strictly depends on the presence of Dish and scube2.

2- The mono lipidated form bind Patch via a protein protein interface together with the N terminal peptide and mediates full Shh activity in vitro and in certain biological context in vivo. Their results also indicate the N-terminal Shh peptide interacts with the

first, but not with the second extracellular domain of Ptch1.

3- Their results also suggest that the N peptide on its own is not sufficient for signaling. . 

Based on these findings, the authors suggest that dual- or monolipidated Shh signaling to Ptch1 is not mutually exclusive, but may be differently used in different tissues and in different developmental contexts thereby explaining functional Hh versatility during development.

Important observations:

Whereas most of the data presented in this paper are of high quality and scientific interest serious revisions have to occur before publication. 

I don’t grab relevance of the paper title

The paper needs serious rewriting to increase clarity and accuracy. 

The Introduction needs to be focusing on questions asked: “Functional versatility of HH proteins partly due to posttranslational modifications”. Specifically, a clear scheme on Shh posttranslational modifications and its maturation process. together with scube 2, Dispatch, AA and lipids involved has to be added to the manuscript. All numbers and statistical analysis shall be removed from the text and put into figures legends.

The author should focus on what they establish and stick to their observations while keeping suggested interpretations to their minimum 

Whereas most of what the author claim is reasonably proved, I have concerns about the last part regarding whether the N peptide on its own is not sufficient for signaling. Indeed, design of the experiment cannot eliminate the possibility that steric hindrance from the mcherry protein affects the validity of their observations (See details below) At minimum this needs to be clearly stated and discussed.

Material and methods

lane 104

What is mbcd? 

Part 3.1

Figure 1 

What is the small red part in the little scheme?

The WB shall be described in the figure legends. Why do you have either one band or two bands? I could not find which antibody they use for the WB.

Part 3.2

The author test differentiation of C3H10T1/2 cell line following expression of specific differentiation markers by RTqPCR

Figure 2

1- In the WB Why 2 bands for Shhc and one for Dual lipidated Shh? 

2- Fig 2A where are the error bars? 

3- The difference between B and C should be indicated in the legends.

Part 3.3

Figure 3

Presentation of qRT-PCR results in various formats (Fig 3B compared with Fig 2A or B) depends on the Figure, which diminishes the coherence of the paper and the easiness of reading.

lane 342: I dont understand this sentence “High C25AShhC bioactivities in the same experimental system (Fig. 3D) suggests that ShhC processing likely occurs upstream of amino acids R34/R35.”

Part 3.4:

This paragraph that needs clarification as despite my efforts I had huge difficulties understanding their conclusions.

There is no clear answer to the first question: "are interactions between N-terminal amino acids of the ligand and Ptch1 receptors conserved?"

There are fussy answers to the second question 2- Do the N terminal part contributes to HH function in vivo as deletion of 6AA 91 a 95 maintain a partial activity whereas deletion of AA after 94 decrease the activity of the protein.

Fig 4 B the text should be bigger

Part 3.5

They authors asked whether isolated, palmitoylated or unpalmitoylated N-terminal Hh peptides are sufficient for Ptch receptor binding and signaling in vivo? 

They replace Shh N terminal part with mcherry which allows Cholesteroylation during mcherry secretion to the cell surface, and generate one constructs carrying N terminal HH peptide palmitoylated Pmcherryc another without N terminal HH peptide.

They observed that when expressed in vivo they were both inactive 

The authors conclude that interactions between the palmitoylated Hh N-peptide and the receptor alone are not sufficient to elicit signaling, and that physical N-peptide linkage with the globular Hh domain is a minimal requirement for the generation of bioactive protein, 

This result is questionable. The author should discuss the possibility that mcherry hindrance independently of the peptide, interferes with biological activity.

Discussion 

What does Pdb structure 60es means? 

Models of Ptch activity modulation as tested in this study. It will be great to have access to the structure of the presented proteins on line  to see them in 3D.

This part is confusing and filled with suggestions that are in part far from what the authors have actually shown. We suggest that the authors stick to discussing their results.

Comments on the Quality of English Language

The quality of the English is good but clarity of the overall paper needs to be improved.

Author Response

We would like to thank reviewers for their time, effort and their constructive criticism. In our resubmitted manuscript, we have addressed all points raised by the reviewers:

Reviewer 2 Report

Comments and Suggestions for Authors

This manuscript by Ehlers et al. carried out some experiments to demonstrate that a non-palmitoylated form of Shh has certain signaling activity and identify that eight amino acids in the N-terminal of Shh are critical for its function. Identifying these forms of Shh is very interesting and will provide information toward the understanding of how Shh functions in vivo. Given the importance of the Hh pathway in development and diseases, the topic is of general interest. However, the manuscript suffers from several deficits that need to be carefully addressed. Some data were over-interpreted, and some interpretations have important caveats. Additional experiments are required to clarify a number of important issues.

Major concerns:

1.     In the context of Shh signaling, palmitoylation of the Shh protein is essential for its proper function. Palmitoylation anchors Shh to the cell membrane and is crucial for its signaling activity. The removal of palmitate groups can affect the membrane association and signaling capability of Shh. If Shh is depalmitoylated, it may lose its membrane association and may have reduced or altered activity compared to the palmitoylated form. The specific consequences would depend on the experimental context and the system being studied. In this study, to put ShhC in the context of Shh signaling, does ShhC induce Ptc1 expression in cells that have Shh responsiveness? Did the authors test Shh responding cell lines, such as the NIH3T3 cell line?

2.     It's important to note that the activity of depalmitoylated Shh may vary depending on the experimental conditions and the specific system used for analysis. Does ShhC occur in a mouse model of Shh signaling? If it is the case, where is ShhC located in the cilium model of Shh signaling?

3.     In experiments with C3H10T1/2 cells, it is a positive finding that ShhC induces differentiation of the C3H10T1/2 cells; however, a lot more experiments need to be performed in order to draw the conclusion that ShhC indeed has activity in terms of inducing Shh target gene expression. Overwhelming amount of ShhC added to the cells could have artifacts to induce cell differentiation. The authors need to put this in the context of endogenous expression of ShhC, otherwise, it could be a non-specific effect.

4.     The finding that N-terminal amino acids contribute to Shh function is very interesting; however, what would be the mechanism(s) for these amino acids to control Shh activity? Do these amino acids contribute to the membrane localization of the protein? Do these amino acids contribute to the binding of potential partners? It is not interesting at all if a truncated protein having less activity was seen.

5.     For the experiment using Drosophila model, two major problems in the experimental design: 1) the authors need to use a weaker Gal4 or the endogenous promoter for the expression of Hh variants. The GMR-Gal4 is very strong and might cause non-specific effects; 2) the authors need to characterize the activity of HhC85S in eye discs collected from larva stage to examine their signaling activity in the context of Hh signal transduction. Ci expression and target genes expression should be examined to determine the activity of Hh variants. The adult eye phonotype could be just phenotypic.

 Minor concerns:

1. Line 13, in the first place where it appears, please clearly define the abbreviation “ShhC”.

2. Line 169, primer sequences should have been provided. And, what are the genes specifically examined in this RT-PCR assay?

Comments on the Quality of English Language

needs editing.

Author Response

(The authors gave the same response as above.)

Reviewer 3 Report

Comments and Suggestions for Authors

Ehlers chacracterised a specific form of Shh and analysed its activity for the activation of the signalling. However, the text is quite difficult to follow because of unusual using of the terms and lacking essential explanations. Below are some of the examples.

1) abstract: “we characterize a selectively depalmitoylated soluble variant that is released from dual-lipidated cellular precursors in serum-presence (ShhC). “ The reviewer cannot understand what they mean here. ShhC is a term to indicate the C-terminal part of Shh protein, and it usually does not indicate the protein they indicate here. Also, what are “cellular precursors”?

2) No explanation for Scube2 and Dispatched – what proteins are they?

3) “bioactive“ needs an explanation.

4) “We avoided published protocols of Shh expression into serum-free media”. Please rephrase.

5) Fig. 1 : How are they sure that the eluted proteins are the forms they indicate on the figure?

6) Fig. 2: “ShhC induces strong in vitro differentiation of C3H10T1/2 cells” This cannot be concluded so. They only checked the Shh-responsive genes and they did not check the “differentiation”. 

7) “biofunctions””bioactive” must be precisely defined.

8) “3.5. Isolated palmitoylated or unpalmitoylated N-terminal peptides not active in vivo” is not a sentence.

9) Please ensure that the protein lipidation is regulated completely same as in drosophila and in vertebrates. Otherwise, the Fig5 experiments do not make sense.

10) “We suggest that serum factors required for ShhC release are most likely lipoproteins…” Serum contains a number of unidentified factors and they cannot conclude so.

Comments on the Quality of English Language

needs to be improved.

Author Response

(The authors gave the same response as above.)
